# Cellular Senescence: Pathogenic Mechanisms in Lung Fibrosis

**DOI:** 10.3390/ijms22126214

**Published:** 2021-06-09

**Authors:** Tanyalak Parimon, Miriam S. Hohmann, Changfu Yao

**Affiliations:** 1Cedars-Sinai Medical Center, Department of Medicine, Women’s Guild Lung Institute, Los Angeles, CA 90048, USA; 2Pulmonary and Critical Care Medicine, Cedars-Sinai Medical Center, Department of Medicine, Los Angeles, CA 90048, USA

**Keywords:** cellular senescence, lung fibrosis, pathogenesis

## Abstract

Pulmonary fibrosis is a chronic and fatal lung disease that significantly impacts the aging population globally. To date, anti-fibrotic, immunosuppressive, and other adjunct therapy demonstrate limited efficacies. Advancing our understanding of the pathogenic mechanisms of lung fibrosis will provide a future path for the cure. Cellular senescence has gained substantial interest in recent decades due to the increased incidence of fibroproliferative lung diseases in the older age group. Furthermore, the pathologic state of cellular senescence that includes maladaptive tissue repair, decreased regeneration, and chronic inflammation resembles key features of progressive lung fibrosis. This review describes regulatory pathways of cellular senescence and discusses the current knowledge on the senescence of critical cellular players of lung fibrosis, including epithelial cells (alveolar type 2 cells, basal cells, etc.), fibroblasts, and immune cells, their phenotypic changes, and the cellular and molecular mechanisms by which these cells contribute to the pathogenesis of pulmonary fibrosis. A few challenges in the field include establishing appropriate in vivo experimental models and identifying senescence-targeted signaling molecules and specific therapies to target senescent cells, known collectively as “senolytic” or “senotherapeutic” agents.

## 1. Introduction

### 1.1. Cellular Senescence and Lung Fibrosis

#### 1.1.1. Cellular Senescence and Aging

The progressive deterioration of tissue function occurs during aging. Consequently, aging results in a greater susceptibility to environmental challenges and is a risk factor for many diseases [1,2,3,4], including cardiovascular disease [5], cancer [6], type 2 diabetes [7], Alzheimer disease [8], and organ fibrosis [9]. Our knowledge of the critical mechanisms of aging remains limited. Cellular senescence, one of the hallmarks of aging [1], is defined as a cellular state of irreversibly arrested proliferation of aged or damaged cells [2]. Senescent cells have distinct phenotypic alterations, including genomic instability, telomere attrition, chromatin remodeling, metabolic reprogramming, increased autophagy, decreased mitophagy, and the implementation of a complex pro-inflammatory secretome [10,11]. Although senescence was first characterized as a state of cell cycle arrest after extensive proliferation, referred to as replicative senescence (RS) [12], this cellular process can be induced by different stimuli, including telomere attrition, DNA damage caused by strong genotoxic stress, such as ionizing radiation/IRIS, topoisomerase inhibitors, and oxidative agents/OSIS [13], chromatin perturbations, and oncogene activation/OIS. These triggers initiate senescent programs dependent on or independent of the activation of p53–p21 and/or p16INK4a–pRB pathways, which causes cell cycle arrest.

#### 1.1.2. Pulmonary Fibrosis

Interstitial lung diseases (ILDs) and pulmonary fibrosis are a heterogeneous group of lung diseases characterized by varying degrees of inflammation and fibrosis of the lung parenchyma. Some of these may occur secondary to a known precipitant such as medications, autoimmune or connective tissue disease, hypersensitivity to inhaled organic antigens, or sarcoidosis, while others have no identifiable cause and are classified as idiopathic interstitial pneumonias (IIPs) [14]. Idiopathic pulmonary fibrosis (IPF) is one of the most aggressive forms of IIP, with a median survival time of 2–3 years after diagnosis [14]. IPF is characterized by chronic, progressive fibrosis associated with an inexorable decline in lung function. It occurs primarily in older individuals, is limited to the lungs, and is defined by the histopathologic and/or radiologic pattern of usual interstitial pneumonia (UIP) [14]. Features of this pulmonary disease include temporal and spatially heterogeneous fibrosis, the presence of fibroblastic foci, and the excessive deposition of disorganized collagen and extracellular matrix (ECM), which result in the loss of normal lung architecture, with or without honeycomb cyst formation [15].

Multiple risk factors have been reported to increase the risk of development of IPF in an independent or coordinated fashion. The endogenous risk factors include genetic background, aging, gender, and pulmonary microbiology, whereas the exogenous factors include cigarette smoking, environmental exposure, and air pollution, especially the dust or organic solvents exposure in the occupational population. Comorbidities such as obstructive sleep apnea, gastroesophageal reflux, and diabetes mellitus are also important factors [16]. Nevertheless, the most prominent risk factor for IPF is aging. A longitudinal study to identify independent risk factors of ILD development found that people over 70 had a 6.9 times higher risk for ILD than those over 40 years old [17]. Accordingly, the prevalence and incidence of IPF increase dramatically with age [18]. Studies using less-restrictive criteria to define IPF report an incidence ranging from 2 to 30 cases per 100,000 person-years and a prevalence of 10–60 cases per 100,000 people (reviewed in [19]). In patients >65 years, the prevalence of IPF is reported to be as high as 400 cases per 100,000 people [18], reinforcing the notion that IPF is an age-related disease. In fact, the biological processes underlying IPF have been described as an aberrant reparative response to repetitive alveolar epithelial injury in a genetically susceptible aging individual [20].

It has been long recognized that cellular senescence contributes to an aging-related decline in tissue regeneration capacity and the pathogenesis of aging-related diseases, including IPF (reviewed in [21]). However, the mechanism whereby senescent cells contribute individually or in a coordinated fashion or both to lung fibrosis remains unclear. Herein, we review the current knowledge on the senescence of the primary cell types in the lung, their phenotypic changes, and the cellular and molecular mechanisms by which these cells contribute to the pathogenesis of pulmonary fibrosis.

## 2. Senescence Regulatory Pathways

The complexity of cellular senescence regulation is well-characterized in many disease models, especially aging [22,23]. The two major molecular signaling cascades are the cell cycle regulatory and the senescence-associated secretory phenotype (SASP)-mediated pathways. As mentioned earlier, multiple internal (genetic abnormalities) and external stressors (oncogene activation, oxidative stress, etc.) are the commonly known trigger factors of cellular senescence (Figure 1). However, not all have been explored in lung fibrosis. Our review discusses only the studied regulatory pathways in each prominent cell type significantly involved in lung fibrosis pathogenesis. We find that each cell type uses at least one of both pathways, but nearly all lung cells use the SASP, suggesting the intimate interconnection of these cells in regulating lung fibrosis through cellular senescence.

### 2.1. Cell Cycle Regulatory Pathway

The regulation of senescence through the cell cycle is mediated through classical cell signaling molecules such as DNA-damage response (DDR), the ARF- and MKK3/MMK6-p38MAPK axis, oncogenic signaling (RAS, MYC, and PI3K), and TGFβ/SMAD. These signaling molecules directly activate cell cycle inhibitors, including p14, p15, p16, p17, p21, and p27, and indirectly through TP53 [22,23]. The cell cycle inhibitors then inhibit cyclin-dependent protein kinase (CDK) and their cyclin partners: CDK1/cyclin B, CDK2/cyclin A/B/E, and CDK4/6/cyclin D [23]. This inhibition prevents the phosphorylation of Rb. Phosphorylated Rb loses its ability to associate with a cell pro-proliferative transcriptional factor, E2F1-3, thereby averting normal cell cycle progression (Figure 1) [22,24,25]. This pathway is well-described in crucial cellular players in lung fibrosis, including alveolar type 2 cells (AT2), lung fibroblasts, and immune cells; further details are discussed in Section 4, “The role of cellular senescence in the pathogenesis of pulmonary fibrosis”. Some common examples are telomere attrition, or the mutation of telomerase reverse transcriptase (TERT) [26] and the loss of the Sin3a gene [27] in AT2 cells cause spontaneous lung fibrosis by means of the p53/p21^WAF1/CIP1^ axis. The same axis was also described in plasminogen activator inhibitor-1 (serpine 1)-induced AT2 senescence [28]. The other miscellaneous triggers include autophagy in the cigarette smoke-induced lung fibrosis model [29], whereby autophagy activates negative feedback on mTOR activity regulated through the Ras/PI3K signaling pathway [30]. The PTEN/Akt axis is involved through Akt activation, secondary to the loss of PTEN, resulting in AT2 senescence [31]. Chronic activation of Wnt/β catenin signaling in AT2 cells promotes senescence by directly inhibiting cyclin-dependent protein kinases [32]. The regulation of cellular senescence cyclin-dependent protein kinases is not specific to AT2, as this has been described in fibroblasts. For instance, hyaluronan synthase 2 (HAS2) regulates lung fibroblast senescence through the p27-CDK2-SKP2 pathway [33].

### 2.2. Senescence-Associated Secretory Phenotype (SASP) Regulatory Pathway

The senescent phenotype is not limited to growth arrest. Senescent cells are metabolically active and undergo widespread changes in the expression and secretion of the protein, ultimately developing the SASP. The SASP has been described in multiple senescent cell types in the lung. They are involved in cellular senescence via autocrine and paracrine function [34,35,36]. The primary regulators of SASPs are the three major transcription factors: TP53, nuclear factor kappa-light-chain-enhancer of activated B cells (NF-κB), and CCAAT/enhancer-binding protein-β (C/EBPβ) (Figure 1) [22,23,24,25]. The prominent SASPs in fibroproliferative lung diseases, which include growth factors such as TGFβ and many inflammatory cytokines/chemokines such as IL-6, TNF-α, and MMPs, mainly regulate senescence through TP53 and NF-κB [37]. SASP regulation is not limited to transcriptional factors. In this case, IL-4 and IL-13, released from senescent AT2, promotes alveolar macrophage activation in lung fibrosis via the PAI-1/TGFβ1 axis [38]. C/EBPβ also plays an important role in experimental lung fibrosis, as demonstrated by C/EBPβ null mice that developed less lung fibrosis after bleomycin injury due to the reduced production of lung pro-inflammatory cytokines TGFβ1, TNF-α, and IL-1β, and myofibroblast differentiation [39]. A more recent study showed less typical SASP, leukotrienes, secreted from senescent fibroblasts, but their common regulator still remains unknown [37].

## 3. The Role of Cellular Senescence in the Pathogenesis of Pulmonary Fibrosis

### 3.1. Epithelial Cells

#### 3.1.1. Epithelial Senescence in IPF

Recent studies have indicated that the initiation and/or progression of IPF are linked to epithelial progenitor cell dysfunction. Accelerating aging, including the loss of epithelial progenitor cell function and/or numbers and cellular senescence, is one of the key factors in IPF pathogenesis [40,41,42]. Furthermore, multiple genome-wide screens for IPF-associated gene variants have defined a serial of gene variants linked to defects in the host defense and regulation of cellular senescence, as well as epithelial cell function [43,44]. In up to 15–20% of the familial IPF cases, causative mutations have been identified in genes involved in regulating telomere function—telomerase reverse transcriptase, TERT; telomerase RNA component, TERC; dyskerin, DKC1; telomere interacting factor 2, TINF2; and regulator of telomere elongation helicase, RTEL1. Mucin 5B, MUC5B promoter polymorphism, which is involved in the regulation of mucus production by goblet cells, is among the top risk variants that have been identified for IPF in multiple studies [43,44,45,46,47]. Mutations in the surfactant protein-C gene (SFTPC); surfactant protein A2, SFTPA2; and ATP-binding cassette member A3, ABCA3, whose product is expressed exclusively by alveolar type 2 (AT2) cells, suggest that defects in AT2 cell function may play critical roles in IPF pathogenesis.

#### 3.1.2. AT2 Cell Cellular Senescence in Pulmonary Fibrosis

The AT2 cell, a facultative stem cell that produces and secretes pulmonary surfactants in its quiescent state, is considered the progenitor cell responsible for normal maintenance of the alveolar epithelium [48]. Genetic or environmental factors can compromise its ability to proliferate and can lead to defective alveolar epithelium maintenance. Recent studies using single-cell RNAseq have shown both the regional depletion of AT2 cells and abnormal activation of cellular senescence and senescence-related signaling pathways such as p53 signaling, mitochondria dysfunction, and oxidative stress, in the fibrotic region of IPF explant tissue [27,49,50], also evidenced by increased SA-β-gal signaling and the expression of p16 and p21 in SFTPC-positive AT2 cells. Different genetic in vivo animal models (Table 1) and in vitro precision-cut lung slides models have been developed to demonstrate AT2 cell dysfunction, especially cellular senescence in the initiation and progression of pulmonary fibrosis.

Cellular replication leads to the gradual shortening of telomeres, which are composed of DNA repeats and associated proteins at the ends of chromosomes. The shortening of telomeres has been tightly linked to the induction of cellular senescence [59]. Replicative senescence cells accumulate in tissue with age- and age-related diseases, including IPF. As indicated by GWAS studies, telomere function-related genes are among the top genetic variants that have been identified for IPF. Animal models focusing on the role of telomere function have also demonstrated the causal relationship of telomere disfunction in AT2 cells for pulmonary fibrosis. Tert and Terc are the two key components of the telomerase holoenzyme [60]. Neither null mutant of Tert (Tert−/−) nor Terc (Terc−/−) mice develop spontaneous lung fibrosis [61,62], even with inbred mice up to the fourth to sixth generation of Tert^−/−^ with the requirement of bleomycin dosage to induce lung fibrosis [51]. The third generation of *Terc*^−/−^ mice showed enhanced fibrosis upon liposaccharide and bleomycin treatment [52]. This might be explained by the fact that the loss of telomerase activity does not lead to the shortening of telomere length immediately by itself [51]. Similarly, mice with AT2-specific Tert deficiency did not develop spontaneous lung fibrosis up to 9 months. However, upon bleomycin injury, they showed enhanced signs of AT2 cellular senescence and lung fibrosis [26]. The shelterin complex, which recognizes the sequence repeats of TTAGGG that are added by telomerase and protects the chromosome ends, is formed by Trf1, Trf2, Tin2, Rap1, Tpp1, and Pot1 [63]. The loss of Trf1 in AT2 cells leads to spontaneous pulmonary fibrosis with the activation of DNA damage, indicated by the increased expression of γH2ax, short telomeres, and accumulation of senescent cells [51,64]. The loss of Trf2 in AT2 cells showed an increased DNA damage response and activation of the p53 pathway. Furthermore, these mice uniformly die after bleomycin treatment, which indicates the essential role of telomere function in AT2 cells [53]. Telomeric repeat-containing RNA (TERRA), a long noncoding RNA (lncRNA) whose overexpression drives the early onset of cellular senescence [54], has been shown as increased in IPF patients [65]. The potential link between TERRA and oxidative stress, via the regulation of mitochondria function, was shown in A549 and MLE12 cell lines, which show some signatures of human or mouse AT2 cells, respectively [65].

Endoplasmic reticulum (ER) is one of the organelles that is responsible for protein biosynthesis and processing. The perturbation of ER homeostasis by stimuli such as mutations, hypoxia, oxidative stress, chaperone shortage, and unfolded or misfolded proteins leads to a stress condition called “ER stress”, which is toxic and detrimental to cells. The unfolded protein response (UPR) is one of the programs that aims to restore ER’s physiological activity—protein homeostasis (proteostasis). ER stress, UPR, and senescence are interconnected. ER stress and activation of the UPR have been observed in senescence induced by different stimuli and have been proposed as the consequence of the senescent phenotype [66]. On the other hand, the augmentation of UPR can also be a driver of senescence [67]. SFTPC mutations are among the top genetic variants of IPF. In a transgenic mouse model, the induced expression of mutant SFTPC^L188Q^ in AT2 cells induces ER stress and increases bleomycin susceptibility [68]; the induced expression of mutant SFTPC variants SFTPC^I73T^ and SFTPC^C121G^ in adult AT2 cells leads to spontaneous pulmonary fibrosis with the activation of ER stress and induction of AT2 cells secretion of SASP-related cytokines [56,57]. The deletion of Grp78, the chaperon protein of UPR, from AT2 cells leads to spontaneous lung fibrosis. Grp78-deficient AT2 cells showed evidence of ER stress, senescence, and impaired progenitor capacity [58].

The excessive production of reactive oxygen species (ROS), which is called oxidative stress, is another critical inducer of cellular senescence [55]. Chronic ROS exposure can lead to oxidative damage to the mitochondria due to the sensitivity of mitochondrial membrane phospholipids to reactive species and can further induce more ROS production [69]. The accumulation of mitochondrial dysfunction leads to the deficiency of oxidizing NADH to NAD and reduces the NAD^+^/NADH ratio, which, in turn, activates AMPK and p53, resulting in a senescent phenotype [70]. The ectopic activation of oxidative stress and the mitochondrial dysfunction signal pathway has been observed in IPF AT2 cells compared to donor AT2 cells [27]. PTEN-induced putative kinase 1 (PINK1), a key mediator of mitophagy, is depleted in fibrotic lungs [71]. The loss of Pink1 in mouse AT2 cells results in mitochondrial dysfunction, upregulation of senescence markers (p16 and p21), and increased levels of TGF-β expression [71,72]. In a cigarette smoke-induced mouse model of lung fibrosis, Zhang et al. showed that cigarette smoke induces AT2 cells senescence through increased mitochondrial ROS and decreased autophagy and mitophagy [29].

The tumor suppressor gene, p53, is one of the master regulators of cellular senescence. The p53 pathway is among the top activated pathways identified in IPF lungs, including AT2 cells, when compared to donor samples. The acetylation of p53 has been shown to play an essential role in the stabilization and activation of p53 in fibrotic lungs [27,50,73,74]. An increased level of acetylation of p53 is observed in IPF AT2 cells, as well as in mouse AT2 cells from bleomycin and TGF-β1-induced fibrotic lungs. The delivery of caveolin-1-derived peptide inhibits the p53 pathway in AT2 cells and reduces fibrosis induced by bleomycin and TGF-β1 [73]. Sirtuin 1 (Sirt1), a NAD-dependent deacetylase sirtuin-1 (a class III histone deacetylase), is one of the p53 deacetylases [75]. A decreased SIRT1 expression level has been observed in IPF [73], and the activation of Sirt1 expression by Sirt1 activator reduces lung fibrosis in the bleomycin-induced lung injury mouse model [76,77]. The conditional loss of Sin3/HDAC complex protein Sin3a in AT2 cells leads to the p53-dependent cellular senescence of AT2 cells and spontaneous lung fibrosis, which mimics the IPF pathogenesis process [27,78]. The loss of Sin3a results in an increased acetylation level of p53 [79], whereas the simultaneous knockout of Sin3a and p53 in AT2 cells reduces lung fibrosis. These data demonstrate that the acetylation of p53 in AT2 cells may play an important role in the activation of the p53 pathway and in driving AT2 cell senescence in pulmonary fibrosis.

#### 3.1.3. Basal Cells

Airway basal and basal-like cells are gaining interest in lung fibrosis pathogenesis as these cells are aberrantly expanding in IPF. This phenomenon is consistent with the “bronchiolization” process that has been observed in end-stage lung fibrosis [80]. The abnormally increased proliferation of airway basal cells such as classic murine Trp63^+^Krt5^+^ and human p63^+^KRT5^+^ cells, human KRT5^+^KRT14^+^p63^+^ [50,81], KRT15^+^ [82], and KRT17^+^ [82,83,84] have emerged in lung repairing processes. Although the proliferation of these basal cells during lung fibrosis argues against the senescent characteristics, two studies indicated that these KRT15^+^ and KRT17^+^ basal cells expressed cellular senescence characteristics at the transcriptomic [82] and protein level [83], suggesting their potential roles in IPF pathogenesis.

More recently, one single-cell RNA-seq cohort identified aberrant basaloid cells in IPF that co-expressed basal epithelial (*TP63*, *KRT17*, *LAMB3*, and *LAMC2*), mesenchymal (*VIM*, *CDH2*, *FN1*, *COL1A1*, *TNC*, and *HMGA2*), and senescence (*CDKN1A*, *CDKN2A*, *CCND1*, *CCND2*, *MDM2*, and GDF15) signature genes [85]. Another study also described aberrant basaloid cells in IPF and systemic sclerosis-related ILDs, whereby those cells expressed senescence markers, *CDKN2A* (gene for p16), *CDKN1A* (gene for p21), and *GDF15* [86], and that cellular senescence was one of the profibrotic regulatory pathways identified in these cells. The new report of “transitional stem cells” or Krt 8^+^ alveolar differentiation intermediate cells displayed a senescent phenotype. p53 and NF-kB activation was abundantly found and persistent in the IPF lungs [87]. Although pathological functions of these basal cell subtypes remain to be explored, accumulating evidence supports their involvement in lung fibrosis through the cellular senescence pathway.

### 3.2. Lung Fibroblasts

Fibroblasts have a pivotal role in wound healing in response to lung injury. Following damage to the epithelium, fibroblasts are activated to proliferate locally and migrate to the sites of injury to rebuild the ECM scaffold for tissue repair. At the site, fibroblasts differentiate into myofibroblasts, which exhibit contractile function and produce ECM components that are essential to wound healing [88]. As the normal repair process progresses, myofibroblasts gradually become senescent, which reduces fibroblast activation, ECM deposition, and limits the progression of fibrosis. In this sense, cellular senescence has an important function in curbing the accumulation of fibrotic tissue and facilitating the resolution of fibrosis [89,90,91]. However, studies showing that targeting senescent fibroblasts and/or myofibroblasts with senolytic drugs significantly ameliorates pulmonary fibrosis and function in mice models of IPF have revealed a deleterious role for these cells in lung fibrosis [41,92]. Indeed, nonresolving bleomycin-induced lung fibrosis in aged mice has been attributed to the persistence of senescent myofibroblasts in the lungs of these mice [93].

Numerous studies have characterized fibroblasts derived from fibrotic lung diseases such as IPF and compared them to age-matched controls. Overall, there is a consensus that fibroblast senescence is increased [93,94] and persistent in the IPF lung [41,93]. IPF lung-derived fibroblasts exhibit several features of cellular senescence. In vitro, these cells exhibit an enlarged flattened morphology, lower cell division rates, and increased expression of SA-β-galactosidase compared to control fibroblasts at the same passage [92,94,95]. IPF lung-derived fibroblasts express increased levels of cell cycle inhibitor proteins p21^waf1^ and p16^ink4D^ [92,94], as well as telomere shortening [94] and the presence of the SASP [41,92,94,95]. In addition to the aforementioned hallmark features of cellular senescence, fibroblasts from IPF lungs also exhibit metabolic reprogramming, mitochondrial dysfunction, insufficient autophagy, and reduced apoptosis [92,94]. Most, if not all, of these characteristics of senescent IPF fibroblasts have been linked to the pathogenesis of the disease.

#### 3.2.1. SASP

Although the SASP plays an important physiological role, such as cell proliferation and differentiation during wound repair, its persistence in the environment may contribute to IPF pathology. Senescent IPF fibroblasts secrete numerous pro-inflammatory cytokines/chemokines, pro-fibrotic factors, and reactive oxygen species [41,92,94,95]. These biologically active molecules can profoundly affect themselves (autocrine effect) or neighboring cells (paracrine effect). It has been reported that the SASP can spread senescence signals to surrounding cells, contribute to persistent inflammation [96] and tissue remodeling [95], and promote profibrotic phenotypic changes of fibroblasts and/or macrophages. The SASP also seems to contribute to immune senescence in IPF. IPF patients have reduced numbers of NK cells in the lung with an impaired phenotype that appears to be controlled by the lung microenvironment through the SASP [97]. Importantly, the deficiency in cytotoxic NK activity could be one of the altered mechanisms perpetrating the accumulation of senescent fibroblasts and other cell types in IPF lungs.

#### 3.2.2. Mitochondrial Dysfunction

Mitochondria are a central hub not only for cellular energy but also for multiple signaling pathways that converge and interact to regulate mitochondrial energetics, biogenesis, ROS production, preservation, and repair of mitochondrial DNA (mtDNA), and mitophagy. Altered mitochondrial homeostasis is found in several cell types in the healthy and diseased aging lung [98]. Cellular senescence and mitochondrial dysfunction are closely linked: cell senescence, more specifically, persistent DDR signaling, directly contributes to senescence-associated mitochondrial dysfunction [99], while mitochondrial dysfunction drives and maintains cell senescence [70]. The combination of senescence and loss of mitochondrial homeostasis seems to have implications in maladaptive responses to cellular stress, increased vulnerability to injury, and the development of pulmonary fibrosis [71,94,100]. In IPF, dysregulation of several of the regulatory mechanisms that control mitochondrial function has been identified in fibroblasts [94,98,101].

An overall decrease in mitochondrial mass and function is present in IPF lung fibroblasts [94,102]. The decline in mitochondrial mass is associated with an abnormality in mitochondrial biogenesis and mitophagy. Mitochondrial biogenesis is the process of producing additional mitochondria and, by extension, cellular energy production capacity. Features of cellular senescence and IPF, for example, telomere shortening and DNA damage, which activates the DNA damage sensor poly [ADP- ribose] polymerase 1 (PARP-1) and p53, feedback to reduce the activation of PPAR coactivators and reduce mitochondrial biogenesis [103]. Impaired mitochondrial biogenesis in IPF potentially creates an imbalance in the production of cellular energy and the demand for it, resulting in mitochondrial dysfunction. NADPH oxidase 4 (Nox4), which is also known to be increased in senescent IFP myofibroblasts, can suppress mitochondrial biogenesis and bioenergetics via Nrf2 and the mitochondrial transcription factor A (TFAM)-dependent pathway [104]. Mitophagy, on the other hand, is an adaptive response that selectively degrades dysfunctional mitochondria following damage or stress. Mitophagy and mitochondrial turnover are important processes in maintaining cellular integrity and health and normal fibroblast function. Selective mitophagy of damaged mitochondria occurs through PINK1-Parkin signaling, with PINK1 acting as a sensor of mitochondrial membrane depolarization and subsequently activating Parkin, which labels the dysfunctional mitochondrion for trafficking to the autophagosome [105]. The impairment of mitophagy mediated by Parkin deficiency in IPF lung fibroblasts has been associated with the increased TGF-β-induced deposition of extracellular matrix [105]. Defects in mitophagy and autophagy result in the increased production of ROS and the activation of platelet-derived growth factor receptor (PDGFR)/mammalian target of rapamycin (mTOR) signaling pathways that enhance the fibroblast-to-myofibroblast transformation. Accordingly, Parkin-deficient mice develop more severe lung fibrosis [105]. PINK1 expression is also reduced in the aging murine lung and biopsies from IPF patients. The fibroblast-to-myofibroblast differentiation mediated by TGF-β1 was characterized by reduced mitophagy and defects in mitochondrial function, at least in part due to PINK1 deficiency [106].

Importantly, as aforementioned, dysfunctional mitochondria produce increased amounts of ROS, which can promote fibroblast proliferation and activation, promoting fibrosis progression [70]. ROS can also damage lung epithelial cells and promote senescence of the surrounding cells [99,107]. Enhanced ROS production can be secondary to the decrease in mitochondrial membrane potential, which could be restored by inhibiting mTOR activity [108]. mTOR activity also affects the signal transduction between mitochondria and the nucleus. Increased nuclear accumulation of the nuclear factor erythroid 2-like 2 (NFE2L2 or Nrf2) increased the expressions of Nrf1 and nuclear-encoded mitochondrial genes (e.g., TFAM), and mitochondrial-encoded genes involved in oxidative phosphorylation [108]. The coordinated expression of mitochondrial and nuclear genes is necessary for maintaining the normal function of mitochondria and homeostatic levels of ROS. NADPH (reduced form of nicotinamide adenine dinucleotide phosphate) oxidase-4 (NOX4) has also been implicated as a mediator of mitochondrial dysfunction in lung fibrosis. NOX4 can repress mitochondrial biogenesis in lung fibroblasts through the direct inhibition of Nrf2 and TFAM [104]. Interestingly, mitochondrial ROS can regulate NOX4 expression in IPF fibroblasts, suggesting crosstalk between NOX4–mitochondria in IPF [109]. This crosstalk has an important role in the pathogenesis of IPF as NOX4 promotes the generation of ROS, TGF-β-induced (myo)fibroblast activation and epithelial cell death, and the apoptotic resistant phenotype of senescence myofibroblasts in IPF [93,109,110].

#### 3.2.3. Autophagy

Autophagy is a process of lysosomal self-degradation that contributes to the maintenance of homeostatic balance between the synthesis, degradation, and recycling of cellular proteins and organelles [111]. Studies suggest that autophagy plays an essential regulatory role in cellular senescence. Indeed, diminished autophagy is observed during aging, and accelerated aging has been attributed to reduced autophagy [112]. Reduced autophagy is observed in aged mice after lung injury [106]. In IPF, autophagy deficiency has been implicated in the senescence of fibroblast [113]. In lung fibroblasts, Beclin1, LC3, and p62 have been described as autophagy-related biomarkers and demonstrate that there is decreased autophagy activity in lung tissues of IPF patients [114,115]. Beclin1, a key regulator of autophagy in IPF lung fibroblasts, is downregulated compared with normal lung fibroblasts [115,116]. Fibroblasts in the fibroblastic foci express both ubiquitin and p62, which is also an indicator of insufficient autophagy [113]. Cellular senescence affects adaptive responses to stress, decreasing autophagy through the activation of mTORC1 in lung fibroblasts. The activation of this pathway also contributes to the resistance to cell death in IPF lung fibroblasts [117].

Autophagy is also involved in the regulation of the activated phenotype of IPF fibroblasts. An aberrant PTEN/Akt/mTOR axis desensitizes IPF fibroblasts from polymerized collagen-driven stress by suppressing autophagic activity, thereby producing a viable IPF fibroblast phenotype on collagen. This suggests that the aberrantly regulated autophagic pathway may play an important role in maintaining a pathological IPF fibroblast phenotype in response to a collagen-rich environment [118]. Deficient autophagy also seems to contribute to the invasive property of IPF fibroblasts [119].

#### 3.2.4. Apoptosis Resistance

In contrast to senescent epithelial cells, fibroblasts from IPF lungs are highly resistant to apoptosis [92,120,121]. Numerous studies have shown that senescent IPF fibroblasts exhibit decreased sensitivity to cytotoxic and pro-apoptotic signals [92,94]. Similarly, fibroblasts from the lungs of aged mice exhibit higher resistance to hydrogen peroxide and TNFα-induced apoptosis [122]. The resistance to stress-induced cell death of senescent fibroblasts results in the persistence of “damaged” cells that would otherwise undergo apoptosis and be cleared from the site of wound repair [93]. Over time, this leads to the accumulation of senescent fibroblasts [41,93], which express high levels of αSMA and secrete excessive amounts of ECM components, promoting the development of fibrosis [95]. Indeed, p16 and p21 expression in fibroblasts are observed within the foci of IPF lung tissue [93]. Further, little to no evidence of apoptosis was observed in α-SMA expressing cells in these areas, confirming the apoptotic resistant phenotype of senescent myofibroblasts in areas of lung fibrosis [93].

Multiple mechanisms have been attributed to the apoptotic-resistant phenotype of senescent fibroblasts and/or myofibroblasts in lung fibrosis. Perhaps the most studied have been the changes in the levels of Bcl-2 family proteins. Reduced pro-apoptotic proteins Bak and Bax, as well as increased anti-apoptotic protein Bcl-2 family proteins, are found in senescent IPF fibroblasts. The accumulation of Bcl-2 family proteins, which include Bcl-2, Bcl-W, and Bcl-XL, contributes to the apoptotic stimuli resistance of senescent cells [116,123]. Intracellular TGF-β1 signaling increases the Bcl-2 protein level by activating JAK2 and STAT3 [124]. The levels of Bcl-2 and Bax protein in fibroblasts seem to be STAT3-dependent as the resistance to apoptosis can be blocked by inhibiting STAT3 signaling [124]. Alterations in the expression of pro-apoptotic and antiapoptotic genes have also been associated with epigenetic modifications, including histone modification and DNA methylation. In contrast to the Bax gene, the acetylation of histone H4K16 (H4K16Ac) is enriched in the Bcl-2 gene, whereas the trimethylation of histone H4K20 (H4K20Me3) is significantly reduced [124]. These site-specific histone modifications lead to active transcription of the Bcl-2 gene, which enhances apoptosis resistance in senescent fibroblasts [124]. The loss of the mitochondrial deacetylase sirtuin 3 (Sirt3) in lung fibroblasts from IPF and the lungs of aged mice has also been implicated in the senescent apoptotic-resistant phenotype of myofibroblasts. The exogenous restoration of Sirt3 in aged mice promotes the activation of the forkhead box transcription factor FoxO3a in fibroblasts, the upregulation of pro-apoptotic proteins Bim, Bad, and Bid, and the recovery of apoptosis susceptibility [125].

Senescent IPF lung fibroblasts are highly resistant to Fas ligand-induced (FasL) and TNF-associated apoptotic ligand-induced (TRAIL) apoptosis. The decreased expression of FasL receptor and Caveolin-1 (Cav-1) protein, combined with increased AKT activity, seem to contribute to this apoptosis-resistant phenotype in these cells [92]. Increased AKT activity is central to various signaling pathways involved in cell survival, and activation of the PI3K/AKT/mTOR pathway decreases autophagy and contributes to apoptosis resistance in IPF lung fibroblasts [117]. Aberrant activation of the PI3K/AKT pathway has been linked to low caveolin-1 expression in IPF fibroblasts. Low caveolin-1 at the plasma membrane creates a membrane microenvironment depleted of PTEN phosphatase activity, favoring sustained PI3K/AKT activation [126]. In contrast, a low activity of PTEN leading to the inactivation of the transcription activator FoxO3a through the PTEN/Akt-dependent pathway and downstream downregulation of caveolin-1 and Fas expression has also been shown [127]. Increased cell surface Fas expression is necessary to sensitize lung fibroblasts to Fas ligation-induced apoptosis. Importantly, activated AKT promotes higher levels of FasL decoy receptor-3 (DcR3), which competitively binds to FasL, protecting IPF fibroblasts from FasL-mediated apoptosis [128]. The role of these pathways in apoptosis resistance in senescent fibroblasts and pulmonary fibrosis is evidenced by the mitigation of bleomycin-induced fibrosis in aged mice by quercetin via a reduction in AKT activation and upregulation of caveolin-1 and Fas levels [92].

The emergence of the senescent and apoptosis-resistant myofibroblast phenotype has also been attributed to an elevated expression of the ROS-generating enzyme Nox4 and an impaired capacity to induce the Nrf2 antioxidant responses [93]. Lung tissues from human subjects with IPF confirmed a high expression of Nox4 in fibroblastic foci, where Nrf2 expression is reduced. The in vivo knockdown of Nox4 and pharmacologic targeting of Nox4 during the persistent phase of lung fibrosis in aged mice reduced Bcl-2 levels and restored the capacity of senescent fibroblasts to undergo apoptosis, permitting fibrosis resolution.

### 3.3. Mesenchymal Progenitor Cells (MPCs)

Recently, a subpopulation of stage-specific embryonic antigen (SSEA4)-expressing cells with the properties of mesenchymal progenitor cells (MPCs) was identified in the lungs of patients with IPF [129]. These cells were identified as the cell-of-origin for fibroblasts comprising the fibrotic reticulum in IPF. Indeed, gene and protein expression profiling of IPF lung mesenchymal progenitors distinguished them from control lung MPCs, with an enrichment of genes associated with disease-relevant ontologies [129,130,131]. These cells are enriched for pro-fibrotic and senescence factors and exhibit a global loss of transcripts encoding for components of various DNA damage response and repair proteins, including DNA-PKcs [130]. Several studies have previously shown the relationship between DNA damage and repair, senescence, and pulmonary fibrosis [132]. The loss of clusterin, for example, can promote disrepair and senescence in fibrotic lungs via the loss of DNA damage response and repair pathways [133]. Accordingly, the chronic pharmacological inhibition of DNA-PKc activity promoted the proliferation of SSEA4^+^ MPCs and increased the expression of senescence-associated markers in cultured lung fibroblasts [130]. Considering that IPF MPCs produce fibroblast progeny manifesting the full spectrum of IPF hallmarks and can establish fibrosis in vivo [129,134,135], it is conceivable that these cells play an important role in the progression of pulmonary fibrosis. It is noteworthy that mediators such as IL-8 and the chemokine CCl28, which have been shown to be increased in the senescent environment, promote the activation and expansion of SSEA4^+^ MPCs and their fibroblast progeny, as well as their fibrogenicity and expression of markers of cellular senescence [135,136]. Thus, a feedforward loop between the senescent environment, the expansion of MPCs and progeny, and the senescent and pathogenic behavior of these cells seems to be present in fibrotic lung diseases such as IPF.

### 3.4. Immune Cells

The link between cellular senescence and immunity is overwhelming [35,137]. Senescent cells can physiologically activate innate and adaptive immune systems to maintain tissue and organ homeostasis. However, the persistent or accumulation of senescent cells can certainly dysregulate the immune systems and transform the organ microenvironment to favor a chronic inflammatory state that, in part, is commonly observed in many age-related conditions, lung fibrosis included [138,139,140,141]. There are two characteristics of the immune system in the cellular senescence processes implicated in lung fibrosis: the dysfunction of the immune system called “immunosenescence”, and the senescence of immune cells per se [142]. Despite controversy in the role of inflammation in lung fibrosis due to the failure of many immunosuppressants in treating IPF [143], persistent chronic inflammation is undoubtedly one of the hallmarks of lung fibroproliferative disorders [143]. Here, we describe two main characteristics of immune dysfunction associated with cellular senescence in lung fibrosis.

#### 3.4.1. Immunosenescence

Immunosenescence is broadly defined as declining immunity with age or prematurely with specific stimuli [144,145,146]. The main characteristics are a low proliferative index of the immune cells and maladaptive responses to triggers and stressors [147,148,149,150]. Although immunosenescence can occur irrespective of immune cell senescing [142], in most cases, it is the main consequence of immune cell senescence. Immunosenescence contributes to lung fibrosis pathogenesis in two ways: first, it promotes an accumulation of senescent cells that are the primary sources of SASP mediators [151,152], and second, it enhances proinflammatory mediators released from immune cells into the lung microenvironment, providing additive effects to SASP from other sources in reverberating inflammation.

Innate immunosenescence

Innate immunity is primarily driven by alveolar macrophages, natural killer (NK) cells, and dendritic cells (DC). The best characterization of innate immunosenescence is the alteration of the immune system in aging irrespective of the senescence of immune cells themselves [153]. Some of these changes contribute to lung fibrosis development [154]. An example of innate immune dysregulation is an increase in immature DC in the BAL and lungs of IPF patients [155,156]. Immature DCs are the source of multiple proinflammatory cytokines in autoimmunity [157]. They also foster a profibrotic milieu in fibrotic lungs of patients with systemic sclerosis [158]. Moreover, they express toll-like receptor 9 (TLR9), which is implicated in the rapid progression of IPF [159]. Lastly, the decrease in matured CD57^+^ cytotoxic NK cells and leukocytes in end-stage IPF is associated with defective immune surveillance necessary for senescent cell clearance, thereby promoting a profibrotic microenvironment [160].

Adaptive immunosenescence

A shift in the T cell and B cell population toward the memory phenotype and the progressive loss of the naïve cells are key features of adaptive immunosenescence [161]. The primary effect of cellular shifting is the maladaptive immune responses that significantly change the lung microenvironment toward a profibrotic state [146,162]. For instance, an increase in Treg cells in aging promotes Th17 cell differentiation and IL-17 production in experimental lung fibrosis [163,164]. Similarly, IL-17A from T helper cells utilizes the IL-17A/IL-17RA axis to mediate rheumatoid arthritis-induced lung fibrosis [165]. The role of B cells in adaptive immunosenescence in lung fibrosis is much less studied.

#### 3.4.2. Senescence of Immune Cells

The characterization of immune cell senescence is well-described in aging. However, this depiction is less straightforward in most lung fibrosis studies. Nonetheless, due to the intertwined nature of aging and lung fibrosis, it is conceivable that aged immune cells regulate lung fibrosis pathogenesis, in part, through the cellular senescence mechanism [150].

Monocytes and macrophages

Traditionally, the phenotypic changes of aged monocytes and macrophages are inferred to the senescent stage of these cells. Aging decreases the lifespans of progenitor cells, alters surface receptors such as TLRs [166], and blunts the M1 macrophages’ polarization toward M2, thus favoring the pro-inflammatory roles [167,168,169]. All of these changes cause a proinflammatory milieu but become ineffective in clearing senescent cells. Monocytes from IPF and ILD patients display diminished TLR1/2 function and phenotypic changes to acquire a pro-inflammatory state [168,170]. The presence of p16*^Ink4a/^*β-galactosidase-positive macrophages in lung fibrosis resemble senescent macrophages seen in aging mice [171], which could explain senescent cell accumulation observed in lung fibrosis such that these profibrotic senescent macrophages lose their capacity to eliminate cytotoxic senescent cells. However, the controversy arises surrounding the specificity of p16*^Ink4a/^*β-galactosidase positivity features, given that these macrophages did not undergo cell cycle arrest, as typically seen in senescent cells [172]. In this sense, the best characterization of profibrotic senescent IPF alveolar macrophages is via apoptotic resistance induced by mitophagy [173]. In this study, upon stimulation, IPF alveolar macrophages accumulated mitochondrial ROS and underwent mitophagy mediated through Akt1 [173]. Damaged mitochondria accumulation in IPF alveolar macrophages was also reported by others, suggesting the link between mitophagy and senescence [174,175]. Lastly, aged macrophages are more hyperresponsive to noxious stimuli through NOD-, LRR-, and pyrin domain-containing protein 3 (NLRP3) inflammasome activation, demonstrating their profibrotic capacity [176]. Together, senescent monocytes/macrophages play essential roles in the aberrant immune response in lung fibroproliferative processes.

T cells

T-cells and their subtypes, such as natural killer T cells (NKT), γδ T cells, T-helper cells (CD4), and cytotoxic T-cells (CTL, CD8), are involved in IPF pathogenesis [162]. The shift in T cell profiles from highly proliferative “naïve” T cells to steady “experienced” or effector memory T cells (CD25^−^CD45RA^−^CD45RO^+^CD127^+^) is the classic finding of T cell senescence in aging [177]. These memory T cells do not have a robust response to stimuli. Still, they continuously secrete a low level of inflammatory cytokines that resemble SASP, promoting a chronic inflammatory state. The standard molecular markers of senescent T cells are CD27, CD28, CD57, and Killer cell Lectin-like Receptor G1 (KLRG1). Specifically, the loss of CD57 or CD28 coincides with the upregulation of p16 and p21 [177,178]. Functionally, the senescent T cells express higher levels of proinflammatory cytokines such as IFN-γ and TNF-α, or cytotoxic mediators such as granzyme B and perforin, which cause tissue damage and further activate other immune cells [179]. Progressive loss of the CD28 surface marker is one of the characteristics of T cell senescence [179]. Indeed, CD28*^null^* cells are found in BAL and lung explant tissue of IPF patients. The first study showed that CD4^+^ CD28*^null^* T-cells in BAL were associated with a poor prognosis of patients with IPF [180]. The enrichment of CD8^+^CD28*^null^* limited the efficacy of immunosuppressants such as dexamethasone in experimental lung fibrosis [181]. Although it is unclear whether targeting senescent T cells will improve clinical outcomes in lung fibrosis, they remain one of the promising therapeutic targets through alternative approaches. For example, PD-1-expressing CD8^+^ T cells augmented lung fibrosis in a humanized mouse lung fibrosis model. Still, in the presence of PD-L1, the engagement of the PD-1/PD-L1 pathway alleviated lung fibrosis through T-cell suppression, suggesting the beneficial effect of PD-L1 [182]. However, PD-L1 in nonimmune cells such as lung fibroblasts has the opposite impact; specifically, it promoted fibroblast invasiveness and augmented lung fibrosis [183].

Other Immune Cells

Additional immune cells worth mentioning are circulating leukocytes, whereby the shortening of the telomere length of these cells was noted in familial and sporadic pulmonary fibrosis patients [184]. As telomere shortening is characteristic of replicative senescence [22], it is conceivable that these “senescent leukocytes” are, at least in part, involved in the aberrant immune response in IPF. More recent evidence (in an abstract form) showed that SASP released by IPF senescent fibroblast promoted the senescence of pulmonary NK cells that, in turn, permitted the accumulation of senescent fibroblasts, thereby creating a vicious cycle of persistent chronic inflammation [97]. Moreover, the decrease in NK cell numbers in IPF patients is associated with an imbalance in the Treg/Th17 axis. Specifically, CD3^+^CD4^+^CD25^high^Foxp-3^+^ cells were elevated in IPF, whereas Th17 cells were significantly compromised [185]. However, some conflicting results on the absolute numbers of NK cells are noted in IPF and ILDs [186,187].

In summary, despite the debate surrounding the contribution of chronic inflammation in lung fibrosis pathogenesis, more direct and indirect evidence supports the roles of the senescent state of the immune response. The intertwined immune dysregulation in cellular senescence is the basic principle of senescence pathogenic mechanisms. In addition, impaired immune surveillance is responsible for the accumulation of senescent cells in aging lungs [188]. The continual accretion of these senescence cells, in turn, elevates SASP mediator levels in the fibrotic foci that can further recruit immune cells, promoting a chronic persistent inflammatory state and the progression of fibrosis [139,189].

### 3.5. Other Cells

#### Bone-Marrow Mesenchymal Stem Cells (B-MSCs)

MSCs, also known as mesenchymal stromal cells, are a population of adult stem cells that were first described in the bone marrow but have been described in many tissues. These cells are multipotent stromal cells than can differentiate into a variety of cell types and, thus, have an important role in tissue remodeling and repair [190]. Indeed, B-MSCs are one of the experimental stem-cell-based treatments in lung fibrosis. In silica-induced lung fibrosis in mice, intravenous administration of B-MSCs ameliorated lung fibrosis in that B-MSCs homed to injured lungs and replenished the injured epithelial cells [191]. Therefore, the senescence of these stem cells would impair the tissue repairing capacity that is needed in IPF. B-MSCs of IPF patients displayed senescence phenotypic changes, including the decreased replication rate, upregulation of common senescence markers, and the presence of DNA damage. These cells were profibrotic, as evidenced by their capacity to accelerate lung fibrosis upon administering to bleomycin-injured mice [192]. 

In summary, known key cellular players in lung fibrosis pathogenesis, i.e., epithelial cells, mesenchymal cells, and immune cells, exhibit cellular senescence phenotypes in pre-clinical studies and in human lung specimens. It further emphasizes the critical role of the senescence regulatory pathway. Although different studies highlight their cells of interest, it is likely that senescence processes are present in more than one cell type in the lung, as has been demonstrated in robust single-cell RNA sequencing studies. It is unclear at this stage whether these senescent cells work in concert to promote lung fibrosis, or whether one dominant cell type drives the process.

## 4. Targeting Senescent Cells for the Treatment of Pulmonary Fibrosis

Due to the detrimental effects of senescence in numerous diseases such as IPF, preventing or disrupting senescence can be beneficial. Recent studies have revealed that targeting senescent cells prevents age-related bone loss in mice [193,194]; senolytic drugs reverse damage caused by senescent cells and improve physical function and increase the lifespan in artificially or naturally aged mice [195]; clearance of the senescent glial cells prevents cognitive decline [196]; targeting senescent oligodendrocyte progenitor cells alleviates cognitive deficits in an Alzheimer’s disease model [197]. In the context of pulmonary fibrosis, targeting senescent AT2 cells using ABT-263, which inhibits Bcl-2/xl, reverses ionizing radiation-induced lung fibrosis [198]; the senolytic drug cocktail dasatinib plus quercetin has been shown to reduce lung fibrosis in multiple lung fibrosis disease models: experimental precision-cut lung fibrosis ex vivo model [42], bleomycin-induced lung fibrosis mouse model [41], and AT2 Sin3a-LOF mouse model [27]. Quercetin alone has also been shown to target senescent fibroblasts and prevent lung fibrosis in bleomycin-induced lung fibrosis in aged mice [93]. Moreover, a small pilot in-human open-label trial of the dasatinib plus quercetin cocktail to 13 IPF patients showed significantly increased physical function measures such as 6 min walk distance, 4 m gait speed, and chair-stands time. However, the reports of pulmonary function, biochemical parameters, and frailty index remained unchanged [199]. Collectively, we can see promising results for targeting cellular senescence for aging-related diseases such as IPF.

## 5. Conclusions

Cellular senescence has emerged as an important driving force in lung fibrosis pathogenesis. Accumulating evidence indicates that senescence processes occur in the critical cellular players involved in lung fibrosis during pathologic lung proliferation. AT2 cells and lung fibroblasts are the main drivers of tissue repair and regeneration after injuries, whereby inflammation also plays essential roles. Some trigger factors and molecular signaling pathways are shared among these cells, for instance, autophagy and mitophagy observed in AT2 and lung fibroblasts. The apparent culprit seems to be the SASP, which causes a vicious loop by proceeding as the trigger and effector molecules. Elimination of senescent cells is a logical concept to counteract the damaging effects of SASP, but this task is far from trivial. The current knowledge gap is to identify how senescent cells in the lung interact, modify the lung microenvironment, and mediate persistent and progressive fibrosis. The advancement and evolution of scientific technologies such as single-cell RNA sequencing or nuclear sequencing and multi-omics approaches will further elucidate the contribution of cellular senescence in lung fibrosis.

## Figures and Tables

**Figure 1 ijms-22-06214-f001:**
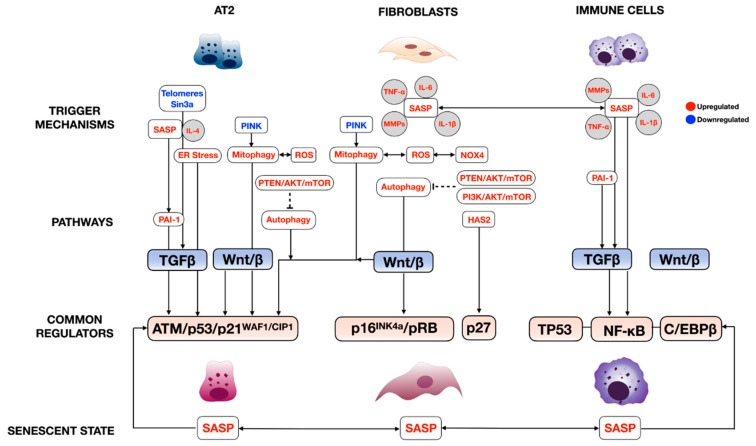
Regulatory molecular signaling pathways of cellular senescence in pulmonary fibrosis. The schematic illustrates the regulatory pathways of cellular senescence in key cellular players involved in lung fibrosis pathogenesis. AT2 = alveolar type 2 epithelial cells; PINK = PTEN-induced putative kinase 1; SASP = senescence-associated secretory phenotype; IL = interleukin; TNF-⍺ = tissue necrotic factor-alpha; ER = endoplasmic reticulum; ROS = reactive oxygen species; NOX4 = NADPH oxidase 4; PAI = plasminogen activator inhibitor; PTEN = phosphatase and tension homolog; mTOR = the mechanistic target of rapamycin; TGFβ = transforming growth factor-beta; ATM = the ataxia telangiectasia mutated; RB = retinoblastoma; PI3K = phosphoinositide 3-kinase; HAS = hyaluronan synthase-2; NF-κB = nuclear factor kappa-light-chain-enhancer of activated B cells; C/EBβ = CCAAT/enhancer-binding protein beta.

**Table 1 ijms-22-06214-t001:** Animal models involved with alveolar type 2 epithelial cell senescence for lung fibrosis.

Animal Models	Spontaneous Fibrosis	Induced Fibrosis	References
**Tert−/−**	No	4th-generation inbred, reduced required dose of bleomycin to induce fibrosis	[51]
**Terc−/−**	No	3rd-generation enhanced fibrosis upon liposaccharide and bleomycin	[52]
Sftpc^CerER^; Tert ^flox/flox^	No	Bleomycin induced enhanced fibrosis	[26]
Sftpc^CerER^; Trf1^flox/flox^	Yes		[51,53]
Sftpc^CerER^; Trf2^flox/flox^	No	Increased susceptibility to bleomycin	[54]
Sftpc^CerER^; Grp78^flox/flox^	Yes		[55]
Sftpc^CerER^; Sin3a^flox/flox^	Yes		[25]
mSFTPC.rtTA; SFTPC^L188Q^	No	Bleomycin induced exaggerated lung fibrosis	[56]
Sftpc^CerER^; SFTPC^I73T^	Yes		[57]
Sftpc^CerER^; SFTPC^C121G^	Yes		[58]

## Data Availability

Not applicable.

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
