# Peer review of "Cellular Senescence: Pathogenic Mechanisms in Lung Fibrosis"

_ijms, 2021, doi:10.3390/ijms22126214_

Round 1

Reviewer 1 Report

In this paper the Authors review the complex molecular mechanisms involved in the pathogenesis of pulmonary fibrosis, with particular emphasis on cell senescence. The review is interesting, well written and up date.

The only issue to be raised  relates to the reference' organization. Many references in fact do not match the corresponding phrases (e.g. ref. 82, 127.131, etc.). Careful re-evaluation of this relevant point are mandatory.

Author Response

We appreciate the reviewer's expertise and time to provide valuable recommendations for us to revise our manuscript. Our response is enclosed in the uploaded PDF file. 

Reviewer 2 Report

The following is a reviewer comment on this review.

  1. The reference numbers do not match throughout the Manuscript. Some of the numbers are missing. Please correct it.

  1. P2 L70-72

Reference is out of date; Raghu G, Remy-Jardin M, Myers JL, Richeldi L, Ryerson CJ, Lederer DJ, et al. Diagnosis of idiopathic pulmonary fibrosis: an official ATS/ERS/. JRS/ALAT clinical practice guideline. Am J Respir Crit Care Med 2018; 198(5):e44-e68. please cite. the IPF diagnostic criteria have also changed and should be revised.

  1. P2 L82-85.

40 years old is correct, not over 40 years old.

  1. P5 L175.

20% of IPF is familial → Clearly wrong. Please correct or delete.

  1. P14 L610-612

CD8+ T cells enhance fibrosis using the PD-1/PD-L1 pathway. → Is this statement correct? Isn't it the other way around?

  1. P4 L129-130

"Dephosphorylated Rb loses its ability to associate with a cell pro-proliferative transcriptional factor, E2F1-3, ... "

This statement is wrong. Phosphorylated Rb loses its ability to associate with E2F1-3. Please correct it.

  1. P5 L177

"telomerase RNA component, TERC; human telomerase RNA component, hTR;"

TERC and hTR are the same. Use TERC and delete hTR.

Author Response

(The authors gave the same response as above.)
